# Frequency and Antibiotic Susceptibility Patterns of *Streptococcus agalactiae* Strains Isolated from Women in Yaounde, Cameroon

Cécile Ingrid Djuikoue [1,*], Paule Dana Djouela Djoulako [2], Rodrigue Kamga Wouambo [2], Rosine Yemetio Foutsa [2], Dorine Ekeu Ngatcheu [1] and Teke Apalata [3,*]

[1] Microbiology Department, Faculty of Health Sciences, Université des Montagnes, Bangangté BP 208, Cameroon
[2] Department of Microbiology and Parasitology, Faculty of Sciences, University of Buea, Buea P.O. Box 63, Cameroon
[3] National Health Laboratory Services, Faculty of Health Sciences, Walter Sisulu University, Nelson Mandela Drive, Mthatha 5117, South Africa
* Correspondence: djuikoe1983@yahoo.fr (C.I.D.); tapalata@wsu.ac.za (T.A.)

**Abstract:** Group B *Streptococcus* (GBS), a commensal in the body, causes a wide range of infectious diseases. This bacterium is dangerous for pregnant women and their babies, in whom it is responsible for early neonatal bacterial sepsis (EOS). The colonisation levels of GBS and its resistance profile to antibiotics provide important information that is useful for orienting prevention strategies. There are few data available on the subject on the determination of resistance phenotypes in Cameroon. We therefore aimed to determine the prevalence of colonisation and antibiotic resistance, including patterns of inducible resistance to clindamycin, of GBS in the city of Yaounde. To achieve this goal, a prospective cross-sectional study with an analytical component was carried out from 28 June to 29 August 2020 at the BIOSANTE laboratory and the Yaounde Gynaeco-Obstetrics and Paediatrics hospital. Vaginal swabs and urine were collected from 163 women. This samples were analysed using 5% defibrinated sheep blood agar and chocolate plus polyvitex agar. The isolates were identified using the morphology of the colony, Gram staining, haemolysis, catalase tests and latex grouping tests. Antibiotic susceptibility testing was carried out by disk diffusion method following the recommendations of the ACFSM 2019. The double disk diffusion method was used to identify isolates with clindamycin-inducible resistance. Our data were analysed with SPSS version 2.1. The results obtained showed that the overall prevalence of colonisation by GBS was 37% (57/163), or 40.3% in non-pregnant women and 59.7% in pregnant women. Pregnancy (*p*-value = 0.019) and earlier (from the second semester of pregnancy) gestational age (*p*-value = 0.025) constituted the risk factors of maternal colonisation by GBS. In addition, the strains of GBS were resistant to all 16 antibiotics tested. A D test showed that 64.7% of GBS strains were constitutively resistant to clindamycin. We also note the presence of M phenotypes. As a whole, our results demonstrated that the rate of GBS colonisation in this study was similar to or higher than those in previous reports in Cameroon. All these results indicate that attention should be paid to this bacterium in the monitoring of antimicrobial resistance and in the care of pregnant women and newborns.

**Keywords:** antibiotic resistance; colonisation; prevalence; GBS; resistance phenotype

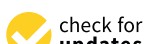



## 1. Introduction

*Streptococcus agalactiae*, equally known as Group B *Streptococcus* (GBS), is a commensal commonly encountered in the human gastrointestinal and urogenital tract [1]. It provokes a broad range of infectious diseases in newborns, elderly people, immunocompromised people, in pregnant women and adults (urinary tract infections) [2]. During pregnancy, about 10% to 30% of women are carriers of the bacterium [3] and 60% of them transmit the

bacterium to their child during pregnancy or childbirth [4]; due to this vertical transmission, the mortality rate from GBS infections in infants has been estimated to be between 2% and 4%, but this rate is even higher in premature infants [5]. Widespread use of intrapartum antibiotic prophylaxis to prevent the early onset of GBS disease has led to concerns about the development of antibiotic resistance among GBS isolates [6]. In order to prevent multiresistance, universal screening of mothers for vaginal or rectal colonisation with GBS between the 35th and 37th gestation week and selective intrapartum antibiotic prophylaxis (IAP) for all women screened positive is the strategy actually recommended to reduce the incidence of colonisation in newborns and to prevent early-onset diseases linked to this bacterium [6]. Despite this prevention, a significant amount of evidence has suggested that GBS has become resistant to antibiotics. A multi-analysis carried out by Mucheye et al. in Africa in 2019 reported resistance to several families of antibiotics [7]; moreover, according to the statistics published by the Center of Disease Control (CDC), the in vitro level of resistance of GBS to erythromycin and to clindamycin has increased by 25–32% and 13–20%, respectively, between 2006 and 2009 in the United States [8]. It is therefore important to determine the prevalence and antimicrobial sensitivity profile of GBS in different regions for the therapeutic strategies. In Cameroon in 2018, Nkembe et al. reported a prevalence of 14% of GBS that was sensitive to β-lactams but had reduced sensitivity to erythromycin [9]. Chatte, in 2014, showed that 7.7% of GBS strains were equally sensitive to β-lactams and macrolides [10].

Due to the presence of antibiotic resistance genes such as *ermB*, *ermTR* and *ermA/E* on transposons, which can travel from one organism to the other [8], the antibiotic resistance of GBS should be studied and monitored regularly. On the basis of the importance of the problem and the questions mentioned above, the present study was carried out with the aim of estimating the prevalence of GBS colonisation in women and antibiotic resistance, and characterising the resistance phenotypes.

## 2. Materials and Methods

### 2.1. Type, Location and Duration of the Study

We carried out a cross-sectional descriptive study with an analytical component in a hospital setting in the central region of Cameroon. The samples were collected and analysed at the Yaounde Gynaeco-Obstetrics and Paediatrics hospital, and the BIOSANTE International laboratory from 29 June to 28 August 2020.

### 2.2. Study Population, Selection Criteria and Sampling Method

We used convenience sampling to recruit pregnant or non-pregnant women who came for consultations or examinations at the collection sites. Women who were in the first trimester of their pregnancy and women on antibiotics or who had been on antibiotics within fewer than 14 days before specimen collection were excluded.

### 2.3. Data and Sample Collection Method

During the consultation or upon the arrival of the women at the laboratory to carry out examinations, we selected eligible patients. The study was explained to them and those who had agreed to participate gave their written consent. Relevant information was collected by means of a questionnaire. Using a cotton swab, we took a vaginal swab and collected urine specimens in labelled tubes from each patient. The collected samples were rapidly forwarded to the laboratory.

### 2.4. Sample Analysis

Culture: Once at the laboratory, all the specimens were inoculated on blood agar supplemented with nalidixic acid–nystatin–colistin (ANC) to inhibit the growth of undesirable bacteria (Gram-negative bacteria, other cocci) and with chocolate plus polyvitex agar, then incubated at 37 °C for 24 h in a jar with 10% carbon dioxide [11].

Identification: Little greyish colonies that were smooth and non-pigmented with a visible beta haemolysis zone appearing after 24 h of incubation were isolated, and their reactivity to the catalase test was evaluated. Lancefield serogrouping was performed on colonies with negative catalase reactivity using the Pastorex-Strep kit to confirm the identification. The identification of other bacterial and fungal species was performed as previously described [12–15].

Antibiogram: The identified GBS isolates were then used to test antibiotics by the Kirby Bauer disk diffusion method [16] on Mueller–Hinton agar plus sheep blood, with the reference strain being *Staphylococcus aureus* ATCC 29213. The antibiotics tested, their disposition and their respective inhibition diameters were those recommended by the Antibiogram Committee of the French Society of Microbiology (ACFSM 2019).

Determination of the resistance phenotype: Determination of clindamycin and erythromycin sensitivity, and determination of the different resistance phenotypes to macrolides, lincosamides and streptogramins b (MLSb were carried out by the double disk diffusion method on Mueller–Hinton agar containing 5% sheep blood. In the case of clindamycin and erythromycin, an area less than 15 mm around the two disks indicated a constitutive resistance to MLBS; the appearance of a D-shaped halo on the medium was considered as indicative of inducible clindamycin resistance. Furthermore, the coincidence of resistance to erythromycin and the absence of resistance to clindamycin was indicative of the M phenotype. Finally, the coincidence of resistance to clindamycin and moderate resistance to erythromycin was indicative of the L phenotype [8,17–19].

### 2.5. Data Analysis and Interpretation

The different variables (age, pregnancy status, gestational age and sample type) and results obtained after verification of their conformity were recorded in Excel 2010 software, then analysed with the statistical software SPSS 21. The principal analyses included calculation of the frequency and the frequency intervals at 95% (for qualitative variables), and the mean or the median (for quantitative variables). The univariate analysis enabled the determination of the factors influencing the occurrence of a GBS infection ($p$-value < 0.05).

### 2.6. Ethical Considerations

Ethical approval was obtained on the basis of the evaluation and validation of the research protocol by the Ethics Committee of the University of Douala. Collection authorisations from the BIOSANTE International Laboratory and the Yaounde Gynaeco-Obstetrics and Paediatrics Hospital were also obtained after validation of the collection request, coupled with the research protocol.

### 2.7. Limitation of the Study

A limitation of this study was the absence of molecular characterisation (genetic identification using 16S Rrna sequencing) of the GBS isolates in this study due to a lack of funding.

## 3. Results

### 3.1. General Characteristics of the Population

In total, 163 patients participated, 90 of whom were pregnant (55.2%) compared with 73 who were not (44.8%) (Table 1). The mean age of the population was 32.3 years ± 10.48, with a minimum of 17 years and a maximum of 73 years of age. GBS was isolated from vaginal swabs only in 84.66% of patients. The prevalence of GBS was 37%. Out of the 57 strains of GBS isolated, 34 were from pregnant women (59.6%) versus only 23 in non-pregnant women (40.3%).

**Table 1.** General characteristics of the population.

| Characteristics | Total Number | Percentage (%) |
|---|---|---|
| Pregnant women | 90 | 55.2 |
| Non-pregnant women | 73 | 44.8 |
| **Age group (in years)** | | |
| 17–25 | 41 | 25.1 |
| 26–34 | 57 | 35 |
| 35–43 | 2 | 19.6 |
| 44–52 | 20 | 12.3 |
| 53 and above | 13 | 8 |
| **Mean age: 32.34 years ± 10.48** | | |
| **Gestational age (in weeks)** | | |
| 27–31 | 41 | 45.6 |
| 32–36 | 22 | 24.4 |
| 37–41 | 27 | 30 |
| **Sample type** | | |
| Vaginal swab | 138 | 84.7 |
| Urine | 25 | 15.3 |
| **Number of GBS strains isolated** | | |
| Pregnant women | 34 | 59.6 |
| Non-pregnant women | 23 | 40.3 |

*3.2. Frequency of Identified Microorganisms*

In total, we found 154 culture positive patients versus 9 culture-negative patients.

It can be seen in Figure 1 that eight different microorganisms were isolated, of which *Streptococcus agalactiae* (37%) was the most prevalent isolate, followed by *Gardnerella vaginalis* (24%), *Candida* species other than *C. albicans* (9.1%), *Candida albicans* (7.8%), *Staphylococcus aureus* (5.2%), *Escherichia coli* and *Klebsiella pneumoniae* (3.9% each), and six (3.9%) co-infections with *Gardnerella vaginalis* and *Candida albicans*.

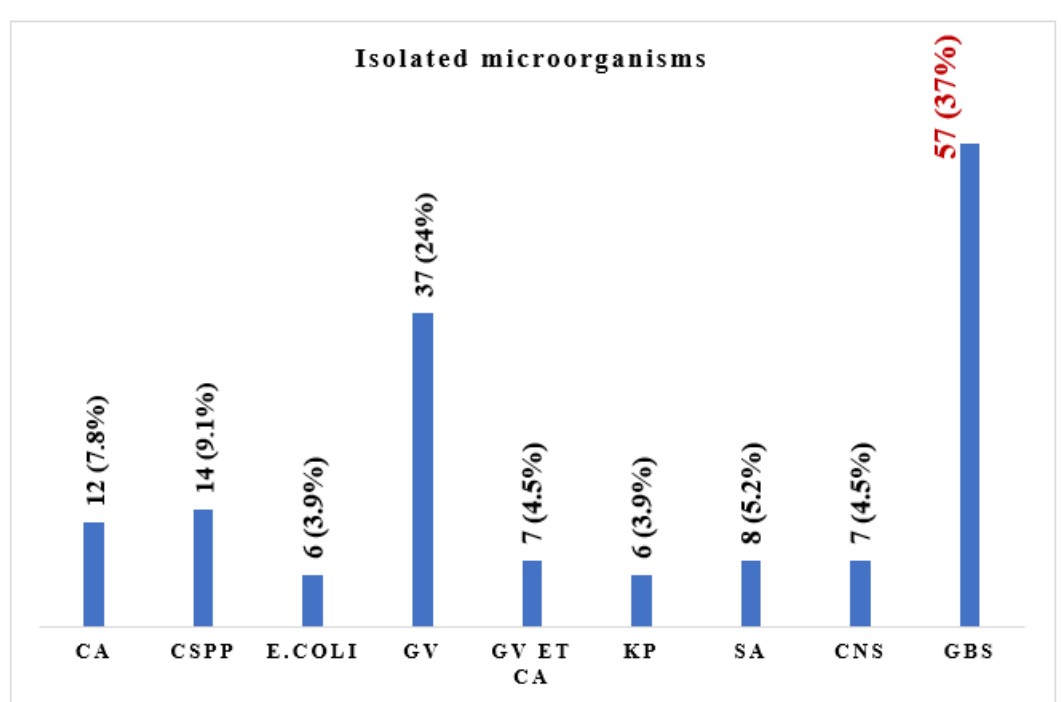

**Figure 1.** Frequency of the different microorganisms isolated.

### 3.3. Antibiotic Resistance Profile of Streptococcus agalactiae

The antibiotics tested are listed in Table 2. Among the antibiotics tested on GBS isolates, penicillin G was the treatment and prophylaxis antibiotic, and erythromycin and clindamycin were alternative treatments and prophylactic antibiotics.

**Table 2.** Antibiotic resistance profile and multiple antibiotic resistance (MAR) index Group B *Streptococcus isolates*.

| Antibiotics | Resistant *n*/57 | Percentages (%) |
| --- | --- | --- |
| Penicillin G | 34 | 58.8% |
| Oxacillin | 34 | 58.8% |
| Amoxicillin | 40 | 70.6% |
| Ceftazidim | 27 | 47.1% |
| Vancomycin | 34 | 58.8% |
| Gentamycin | 27 | 47.1% |
| Streptomycin | 50 | 88.2% |
| Erythromycin | 27 | 47.1% |
| Clindamycin | 37 | 64.7% |
| Tetracycline | 50 | 88.2% |
| Doxycycline | 57 | 100% |
| Chloramphenicol | 50 | 88.2% |
| Norfloxacin | 17 | 29.4% |
| Levofloxacin | 10 | 17.6% |
| Cotrimoxazol | 57 | 100% |
| Bacitracin | 57 | 100% |
| MAR index | Number of isolates/57 | |
| 00 | 08 | 14% |
| 0.1 | 22 | 38.6% |
| 0.2 | 13 | 22.8% |
| 0.3 | 09 | 15.8% |
| 0.5 | 03 | 5.3% |
| 0.7 | 02 | 3.5% |

MAR: Multiple antibiotic resistance (MAR) index.

Of the 57 strains of GBS isolated, all were resistant to one or more than one antibiotic (Table 2). GBS isolates showed the highest rate of resistance to doxycycline (100%) and tetracycline (88.2%). The fewest GBS were resistant to fluoroquinolones (levofloxacin, 17.6%; norfloxacin, 29.4%). The multiple antibiotic resistance (MAR) index for different isolates of GBS revealed 43 (75.44%) with a MAR index less than or equal to 0.2 and 14 (24.56%) greater than 0.2.

### 3.4. Resistance Phenotypes

We noted that 33.33% of GBS were resistant to macrolide–lincosamide –streptogramin b (cMLSbtype) and 19.3% were M type 162. The synthesis of penicillin binding protein (PBP) (31.6%) and 163 synergies were observed in the antibiogram (Figure 2).

### 3.5. Risk Factors Associated with GBS Infection in Women

Our analyses showed that 39% of pregnant women with a gestational age between 27 and 31 weeks were GBS carriers compared with only 31.8% for those between 32 and 36 weeks, and 40.7% for those between 37 and 41 weeks of pregnancy. This difference was statistically significant (*p*-value = 0.025).

The prevalence of GBS was significantly higher in pregnant than in non-pregnant women (*p*-value = 0.019) (Table 3).

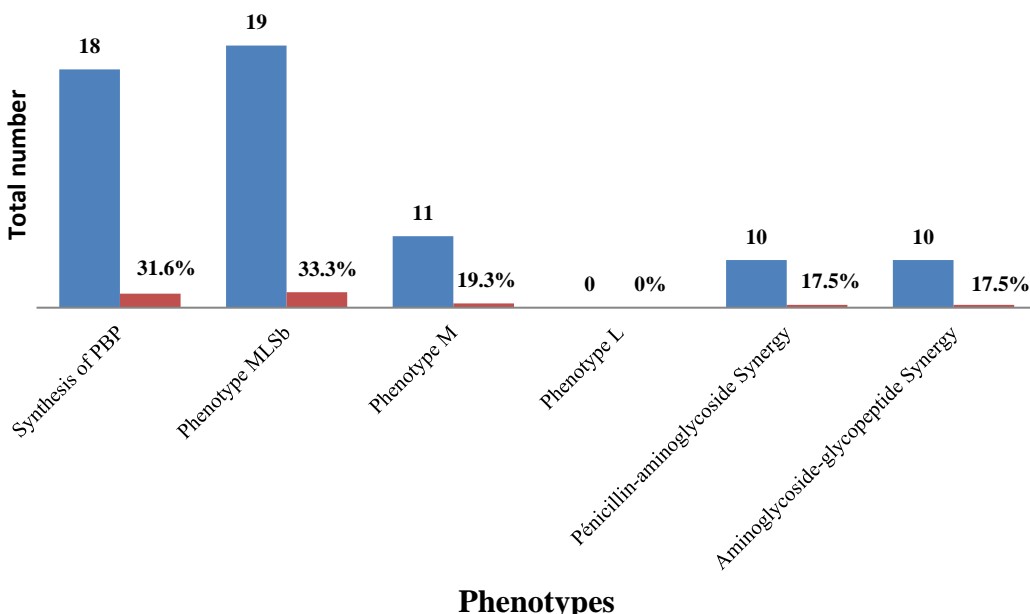

**Figure 2.** Resistance phenotypes and synergies.

**Table 3.** Risk factors of GBS infection in women.

| Variable | | GBS | | Total n (%) | *p*-Value |
|---|---|---|---|---|---|
| | | No | Yes | | |
| Pregnancy | No | 50 (94.3%) | 23 (40.3%) | 73 (44.8%) | 0.019 |
| | Yes | 56 (80.0%) | 34 (59.6%) | 90 (55.2%) | |
| Gestational age (in weeks) | 27–31 | 25 (73.5%) | 16 (39.0%) | 41 (45.6%) | 0.025 |
| | 32–36 | 15 (93.8%) | 7 (31.8%) | 22 (24.4%) | |
| | 37–41 | 16 (80.0%) | 11 (40.7%) | 27 (30.0%) | |
| Age (in years) | 17–25 | 27 (81.8%) | 17 (41.5%) | 41 (25.1%) | 0.890 |
| | 26–34 | 43 (87.7%) | 14 (24.6%) | 57 (35%) | |
| | 35–43 | 21 (87.5%) | 11 (34.4%) | 32 (19.6%) | |
| | 44–52 | 11 (91.7%) | 9 (45.0%) | 20 (12.3%) | |
| | 53 and above | 4 (80.0%) | 9 (69.2%) | 13 (8.0%) | |

*p*-value is significant if <0.05.

## 4. Discussion

The objective of our study was to determine the frequency and sensitivity of GBS. In total, 163 patients were recruited and all samples collected were inoculated on specific media, and GBS was identified by the latex agglutination test.

The most represented microorganism in our study was GBS (37%), followed by *Gardnerella vaginalis* (24%), *Candida spp.* Other than *albicans* (9.1%) and *Candida albicans* (7.8%). These results differ from those reported by Chatté et al. in 2014, where Candida albicans (45.2%) was the most common isolate [10], and those of Nkembe et al. in 2018, where *Candida albicans*, *Gardnerella vaginalis* and *Candida spp.* Had the highest rates of 22%, 2% and 16%, respectively [9]. This can be explained by improved intimate hygiene but an increase in sexual partners, resulting in an increase in the prevalence of certain germs.

The prevalence of GBS was 37%, which is much higher than that reported by Nkembe et al. (2018) in Cameroon (14%) [9], Ali in Ethiopia (13.2%) [20] and Vinnemier in Ghana (19.1%) [21]. Furthermore, the carriage rate of GBS was 40.3% and 59.6%among non-pregnant women and pregnant women, respectively. A higher rate of GBS among pregnant women has been already reported in Ethiopia (25.5%) [22] and Cameroon (21.2%) [23]. This difference can be explained by the fact that the prevalence of GBS infection varies according to regions, and this high prevalence of GBS in pregnant women testifies to its evolution

worldwide. In fact, in comparison with other germs studied, GBS is the most dangerous for the mother and especially for the newborn. Indeed, GBS is the germ most incriminated in early neonatal bacterial sepsis (EOS). In 2020, a study by Liu et al. revealed that GBS was the second germ (after *Micrococcus luteus*) responsible for early-onset bacterial meningitis (ENBM) in full-term infants [11].

The resistance of *Streptococcus agalactiae* to penicillin, amoxicillin, vancomycin, clindamycin and other tested antibiotics was observed in the current study. This observation may help to alert physicians to minimise empirical therapy and establish antimicrobial management in the study area. Similarly, penicillin resistance patterns are different among studies. One study in Cameroon found 100% resistance to penicillin [23], while others in the same country showed no resistance to penicillin, ampicillin and/or vancomycin [9,10]. This can be explained by the fact that the phenomenon of antibiotic resistance is expanding because antibiotics are nowadays used in sectors of activity such as livestock, agriculture, etc.

GBS isolates showed high resistance to erythromycin and clindamycin in our study. This could reduce the possibility of prophylaxis in pregnant women who are allergic to penicillin. One study in Cameroon showed that all GBS strains were sensitive to these antibiotics, except for one (6%) which had intermediate sensitivity to erythromycin [9]. A study in China reported 69.4% and 47.2% resistance to erythromycin and clindamycin [24]; another one in Ethiopia reported 26.5% and 21.4% [22]. These reports are in accordance with the results of our study. In fact, the mechanism governing this resistance relies on the presence of genes such as *ermB*, *ermTR*, *mefA/E* and other antibiotic resistance genes on plasmids and/or transposons. These genes can pass from organism to organism, and monitoring of the antibiotic resistance of GBS should occur regularly [25].

The highest rate of antibiotic resistance was observed for the cyclin family in our study (88.2% and 100% resistance for tetracycline and doxycycline respectively). Similar resistance rates for tetracycline were reported in Ethiopia (73.4%) [22] and Tunisia (97.3%) [26]. The resistance rates to tetracycline, ceftriaxone, erythromycin and clindamycin observed in our study could be due to the widespread use of these antibiotics for different clinical cases, which could lead to the emergence of antibiotic-resistant GBS.

A phenotypic analysis of the 57 GBS that were resistant/intermediate to erythromycin and/or clindamycin in our study revealed that 33.3% of them had Cmlsb phenotypes and 19.3% had M phenotypes. Among the erythromycin- and/or clindamycin-resistant isolates analysed in Ethiopia, 30.6% had an L phenotype, 28.3% of them M phenotypes, 26.1% of them had Cmlsb and 15.2% of them had Imlsb [22]. A Tunisian report also showed that among the erythromycin-resistant isolates, 78.7%, 10% and 2.2% had the Cmlsb, Imlsb and M-phenotypes, respectively [26]. A study in the United States found that 8% of patients had a positive D-test, indicating inducible resistance to clindamycin [27], and another study in Iran demonstrated a rate of 100% for the M phenotype [28]. It is believed that differences in antibiotic use, the practice of prophylaxis, the widespread and indiscriminate use of antibiotics in various clinical cases, variations in susceptibility testing methods and/or disparities in the distribution of serotypes may lead to regional differences in the resistance rates of GBS to different antibiotics.

Knowledge of the risk factors associated with maternal colonisation is useful in reducing the morbidity and mortality of GBS diseases. This study has shown that pregnancy (*p*-value= 0.019) and earlier (from the second semester of pregnancy) gestational age (*p*-value= 0.025) are associated with GBS infection in women. A cohort study including 77 pregnant women in Denmark reported a probable strong influence of gestational status on GBS infection and an increased prevalence of GBS carriage from the first to third trimester but, unlike us, the authors showed an association with the age of the patient [29].

## 5. Conclusions

The present study showed a higher prevalence of maternal GBS colonisation compared with previous Cameroonian studies. Pregnancy and gestational age were found to be the risk factors for maternal colonisation. GBS was found to be resistant to peni-

cillin, amoxicillin, vancomycin, ceftazidime and other antimicrobials tested. GBS with constitutive resistance to clindamycin was identified. In addition, GBS with M phenotypes was detected. Based on these results, GBS screening of pregnant women at the end of the third trimester of pregnancy, pre-prescription antibiotic susceptibility testing, intrapartum antibiotic prophylaxis and large-scale epidemiological studies should be implemented in the study area.

**Author Contributions:** C.I.D. conceived the project and designed the study. C.I.D. searched relevant literature, scrutinised all relevant information and draft the manuscript. C.I.D. conducted and coordinated the field study. P.D.D.D., R.Y.F. and D.E.N., collected and processed the samples and data. P.D.D.D., R.K.W. and R.Y.F. analysed the data and wrote the article. All authors provided additional information. R.K.W. and P.D.D.D. further analysed the data. C.I.D., P.D.D.D. and R.K.W. and T.A. revised the manuscript. All authors have read and agreed to the published version of the manuscript.

**Funding:** This research received no external funding.

**Institutional Review Board Statement:** The study was conducted in accordance with the Declaration of Helsinki and approved by the Institutional Human Health Research ethics committee of the Yaoundé Gynaeco-Obstetric and Paediatric Hospital (Authorisation No. 1124/CIERSH/DM/2020).

**Informed Consent Statement:** Informed consent was obtained from all subjects involved in the study.

**Data Availability Statement:** All data generated or analysed in the course of this study are included in this manuscript.

**Acknowledgments:** The authors would like to thank the staff of the BIOSANTE Laboratory and the Yaoundé Gynaeco-Obstetric and Paediatric Hospital for their financial and material support. The authors are also grateful to the data collectors and participants in the study.

**Conflicts of Interest:** The authors declare no conflict of interest.

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
