# Peer review of "Frequency and Antibiotic Susceptibility Patterns of Streptococcus agalactiae Strains Isolated from Women in Yaounde, Cameroon"

_2036-7481, doi:10.3390/microbiolres13040068_

Round 1

Reviewer 1 Report

This study has investigated the GBS colonisation prevalence and antibiotics susceptibility in a group of pregnant and non-pregnant women admitting to clinic for gynaecologic examination during August 2020. They have cultured vaginal swabs and/or urine samples from 163 patients to isolate GBS and other vaginal microorganism present in the samples and tested GBS isolates for antibiotics susceptibility using disk diffusion method. A prevalence of 37% GBS colonisation was found with higher percentage in pregnant (60%) compared to non-pregnant women (40%). The isolated GBS were highly resistant to most of the antibiotics tested.

Please see my comments below referring to line numbers. Some are just some minor changes/suggestions to make it clearer and some need you to review your data and /or references to make them right.

Thanks for putting these results together and I wish you good luck in publishing your article.

Line 60-61: Are the percentages relate to each of Erythromycin and Clindamycin respectively or both of them combine in 2006 and 2009? The sentence is not clear.

Line 63-64: The prevalence of GBS was 14% or the prevalence of sensitivity to β-lactamines?  The same with line 65? You may say: Chatte in 2014 found 7.7% of GBS were sensitive to…

Line 79: …to recruit pregnant or non-pregnant women who came to…

Line 81: … until less than 14 days before specimen collection…

Line 86-89: Using a cotton swab, we took a vaginal swab and collected urine specimens in labelled tube from each patient. The collected samples were rapidly forwarded to the laboratory

Line 92: …chocolate plus polyvitex…

Line 95: … hours of incubation were selected and their reactivity to catalase evaluated. Lancefield serogrouping was performed on Colonies with negative catalase reactivity using the Pastorex-Strep kit to confirm the identification.

Line 99: …The identified GBS isolates were then used to test antibiotic

Table 1: Effectif (n), please change to Total number

                 Vaginal secretions please change to vaginal swab

Line 134: a minimum of 17 years and a maximum of 73 years of age.

Line 134-135: The GBS was isolated only in the vagina from vaginal secretions representing 88.66% of the patient samples giving us a prevalence of 37%.? What do you mean?

You may change it to: GBS was isolated from vaginal swab only in 84.66% of patients (in the table you have recorded 84.66%?)

I also suggest rounding up the percentages as it’s easier to read and compare.

Line 138: .2. Frequency of GBS strains I suggest change to Frequency of identified microorganisms

You need to mention if you had some culture negative patients. Your total number of patients is 163 but the total number of different isolates in figure 1 is 154. Also is candida species other than candida albicans? If so, you may call it candida spp other than albicans?

 Line 143-147: It emerges from the figure 1 that eight different microorganisms were isolated of which Streptococcus agalactiae (37%) was the most prevalent isolate followed by Gardnerella vaginalis (24.03%), Candida species (9.09%), Candida albicans (7.79%), Staphylococcus aureus (5.19%), Escherichia coli, and Klebsiella pneumoniae 3.9% each and six (3.9%) co-infections with Gardnerella vaginalis and Candida albicans.

Are Candida albicans counted as part of candida species? Are they overlapped?

Table 2: Please change Effectifs (n)/57 to “Resistant N/57”

Line 151: From 57 strains of GBS isolates, all were resistant to one or more than one antibiotic.

GBS isolates showed the highest rate of resistance to doxycycline (100%,) and tetracycline (88.2%). The least number of GBS were resistant to fluoroquinolones (levofloxacin, 17.6% and norfloxacin, 29.4%).

Line 161: We note that 33.33% of GBS were resistant to Macrolide-Lincosamide-Streptogramine B (cMLSB type) and 19.3% were M type 162 (the synthesis of Penicillin Binding Protein (31.58%) and synergies were 163 observed from the antibiogram (Figure 2).

Line 167: Our analyses showed that 39.02% of pregnant women with a gestational age between 27-31 weeks were GBS carriers compared to only 31.82% for those between 32-36 weeks and 40.74% for those between 37-41 weeks of pregnancy. This difference 169 was statistically significant (P-value=0.025)

Which groups are you comparing together? 39% and 40.74% are very close rates. Is this P- value between 39% and 31%? The highest rate belongs to 17-31- and 37-41-week gestation group. Also, in the table it’s not clear p-value refers to which figures. You need to clearly say which group had higher prevalence of GBS compared to other groups like the sentence below:

Line 171: The prevalence of GBS was significantly higher in pregnant than in non-pregnant women (P-value=0.019) (Table 3).

Line 178: The objective of our study was to determine the frequency and antibiotic susceptibility of GBS

Line 180: …specific media and GBS was identified by the latex agglutination test

Line 181: The most represented microorganism in our study was GBS (37%) followed by Gardnerella vaginalis (24.03%), Candida spp. (9.09%) and Candida albicans (7.79%).

Line 183: These results differ from those reported by Chatté et al, in 2014 where Candida albicans (45.16%) was the most common isolate [10] and those of Nkembe et al in 2018 where Candida albicans, Gardnerella vaginalis and Candida spp. had the highest rate, 22%, 18% and 16% respectively [9].

Line 189: The prevalence of GBS was 37% which is consistent with that reported by Nkembe et al, 2018 in Cameroon, and Ali in Ethiopia who had 14% and 13.2% respectively [9, 12] and slightly lower than that of vinnemier in Ghana at 19.1% [13].

Yours is not consistent with those studies, you have much higher rate of GBS than all 3 of them! You have already mentioned that your data is different to Nkembe (see line 183)

Line 199: …may help to alert physicians to minimise empirical therapy and establish…

Line 200: The penicillin resistance patterns are different between studies, one study in Cameroon found 100% resistance to penicillin [15] while others in the same country showed no resistance to penicillin, ampicillin and/or vancomycin [9, 10]. This can be explained by the fact that the phenomenon of antibiotic resistance is expanding because antibiotics are nowadays used in all sectors of activity like livestock, agriculture, etc.

Line 207:…pregnant women who are allergic to penicillin. 207

Line 208: one study in Cameroon showed that all GBS strains were sensitive to these antibiotics, except one (6%) which was intermediate to erythromycin [9]

Line 210: and another one in Ethiopia reported 26.5% and 210 21.4% [14]. These reports are in accordance with the results of our study. 2

Line 232-235: This study showed that pregnancy (P-value= 0.019) and gestational age (earlier gestational age or later gestional age?) (P-value= 0.025) are associated with maternal colonisation.  as reported in other studies, for example in Tunisia only [20], the latter also showed an association with age, which was not the case in our study.

It’s not clear what you are writing here. Is this association between gestational age and GBS colonisation or patient age and GBS? Is it associated with earlier gestational age or younger patients or the other way around? Also as mentioned in your results section it’s not clear which gestational age group are more likely to have GBS. Please clarify.

Line 224-226: A study in the United States found that 8% of patients had a positive D-test, indicating inducible resistance to clindamycin [18] and another study in Germany demonstrated 100% resistance to the M-phenotype [19].

None of these references are in your reference list.

What do you mean by: resistance to the M-phenotype? Please modify your sentence.

Line 234: Tunisia only [20], the latter also showed an association with age, which was not the case in our study.

It’s not clear what you mean here, please clarify.

Author Response

Dear Reviewer #1

Thank you for the comments raised. We are submitting our responses.

Reviewer #1

Comments and Suggestions for Authors

This study has investigated the GBS colonisation prevalence and antibiotics susceptibility in a group of pregnant and non-pregnant women admitting to clinic for gynaecologic examination during August 2020. They have cultured vaginal swabs and/or urine samples from 163 patients to isolate GBS and other vaginal microorganism present in the samples and tested GBS isolates for antibiotics susceptibility using disk diffusion method. A prevalence of 37% GBS colonisation was found with higher percentage in pregnant (60%) compared to non-pregnant women (40%). The isolated GBS were highly resistant to most of the antibiotics tested.

Please see my comments below referring to line numbers. Some are just some minor changes/suggestions to make it clearer and some need you to review your data and /or references to make them right.

Thanks for putting these results together and I wish you good luck in publishing your article.

Line 60-61: Are the percentages relate to each of Erythromycin and Clindamycin respectively or both of them combine in 2006 and 2009? The sentence is not clear.

Authors: Thank you dear reviewer for this remark. We revised the sentence. The percentages relate to each of Erythromycin and Clindamycin respectively

Line 63-64: The prevalence of GBS was 14% or the prevalence of sensitivity to β-lactamines?  The same with line 65? You may say: Chatte in 2014 found 7.7% of GBS were sensitive to…

Authors: Thank you dear reviewer. We revised it

Line 79: …to recruit pregnant or non-pregnant women who came to…

Authors: Thank you dear reviewer. We did it

Line 81: … until less than 14 days before specimen collection…

Authors: Thank you dear reviewer. We did it

Line 86-89: Using a cotton swab, we took a vaginal swab and collected urine specimens in labelled tube from each patient. The collected samples were rapidly forwarded to the laboratory

Authors: Thank you dear reviewer. We did it

Line 92: …chocolate plus polyvitex…

Authors: Thank you dear reviewer. We did it

Line 95: … hours of incubation were selected and their reactivity to catalase evaluated. Lancefield serogrouping was performed on Colonies with negative catalase reactivity using the Pastorex-Strep kit to confirm the identification.

Authors: Thank you dear reviewer. We did it

Line 99: …The identified GBS isolates were then used to test antibiotic

Authors: Thank you dear reviewer. We did it

Table 1: Effectif (n), please change to Total number Vaginal secretions please change to vaginal swab

Authors: Thank you dear reviewer. We did it

Line 134: a minimum of 17 years and a maximum of 73 years of age.

Authors: Thank you dear reviewer. We did it

Line 134-135: The GBS was isolated only in the vagina from vaginal secretions representing 88.66% of the patient samples giving us a prevalence of 37%.? What do you mean?

Authors: Thank you dear reviewer. We meant that: vaginal swab was the most common and was the sample type from which all GBS strains were isolated (84.66%). The prevalence of GBS was 37%.

You may change it to: GBS was isolated from vaginal swab only in 84.66% of patients (in the table you have recorded 84.66%?)

Authors: Thank you dear reviewer. We did it

I also suggest rounding up the percentages as it’s easier to read and compare.

Authors: Thank you dear reviewer. We did it

Line 138: .2. Frequency of GBS strains I suggest change to Frequency of identified microorganisms

Authors: Thank you dear reviewer. We did it

You need to mention if you had some culture negative patients. Your total number of patients is 163 but the total number of different isolates in figure 1 is 154. Also is candida species other than candida albicans? If so, you may call it candida spp other than albicans?

Authors: Thank you dear reviewer. We revised it. We had 9 culture negative patients

 Line 143-147: It emerges from the figure 1 that eight different microorganisms were isolated of which Streptococcus agalactiae (37%) was the most prevalent isolate followed by Gardnerella vaginalis (24.03%), Candida species (9.09%), Candida albicans (7.79%), Staphylococcus aureus (5.19%), Escherichia coli, and Klebsiella pneumoniae 3.9% each and six (3.9%) co-infections with Gardnerella vaginalis and Candida albicans.

Authors: Thank you dear reviewer. We did it

Are Candida albicans counted as part of candida species? Are they overlapped?

Authors: Thank you dear reviewer. Candida albicans are counted on the one hand and other Candida species on the other hand

Table 2: Please change Effectifs (n)/57 to “Resistant N/57”

Authors: Thank you dear reviewer. We did it

Line 151: From 57 strains of GBS isolates, all were resistant to one or more than one antibiotic.

GBS isolates showed the highest rate of resistance to doxycycline (100%,) and tetracycline (88.2%). The least number of GBS were resistant to fluoroquinolones (levofloxacin, 17.6% and norfloxacin, 29.4%).

Authors: Thank you dear reviewer. We did it

Line 161: We note that 33.33% of GBS were resistant to Macrolide-Lincosamide-Streptogramine B (cMLSB type) and 19.3% were M type 162 (the synthesis of Penicillin Binding Protein (31.58%) and synergies were 163 observed from the antibiogram (Figure 2).

Authors: Thank you dear reviewer. We did it

Line 167: Our analyses showed that 39.02% of pregnant women with a gestational age between 27-31 weeks were GBS carriers compared to only 31.82% for those between 32-36 weeks and 40.74% for those between 37-41 weeks of pregnancy. This difference 169 was statistically significant (P-value=0.025)

Authors: Thank you dear reviewer. We did it

Which groups are you comparing together? 39% and 40.74% are very close rates. Is this P- value between 39% and 31%? The highest rate belongs to 17-31- and 37-41-week gestation group. Also, in the table it’s not clear p-value refers to which figures. You need to clearly say which group had higher prevalence of GBS compared to other groups like the sentence below:

Line 171: The prevalence of GBS was significantly higher in pregnant than in non-pregnant women (P-value=0.019) (Table 3).

Authors: Thank you dear reviewer. We revised it

Line 178: The objective of our study was to determine the frequency and antibiotic susceptibility of GBS

Authors: Thank you dear reviewer. We did it

Line 180: …specific media and GBS was identified by the latex agglutination test

Authors: Thank you dear reviewer. We did it

Line 181: The most represented microorganism in our study was GBS (37%) followed by Gardnerella vaginalis (24.03%), Candida spp. (9.09%) and Candida albicans (7.79%).

Authors: Thank you dear reviewer. We did it

Line 183: These results differ from those reported by Chatté et al, in 2014 where Candida albicans (45.16%) was the most common isolate [10] and those of Nkembe et al in 2018 where Candida albicans, Gardnerella vaginalis and Candida spp. had the highest rate, 22%, 18% and 16% respectively [9].

Authors: Thank you dear reviewer. We did it

Line 189: The prevalence of GBS was 37% which is consistent with that reported by Nkembe et al, 2018 in Cameroon, and Ali in Ethiopia who had 14% and 13.2% respectively [9, 12] and slightly lower than that of vinnemier in Ghana at 19.1% [13].

Yours is not consistent with those studies, you have much higher rate of GBS than all 3 of them! You have already mentioned that your data is different to Nkembe (see line 183)

Authors: Thank you dear reviewer. We revised it

Line 199: …may help to alert physicians to minimise empirical therapy and establish…

Authors: Thank you dear reviewer. We did it

Line 200: The penicillin resistance patterns are different between studies, one study in Cameroon found 100% resistance to penicillin [15] while others in the same country showed no resistance to penicillin, ampicillin and/or vancomycin [9, 10]. This can be explained by the fact that the phenomenon of antibiotic resistance is expanding because antibiotics are nowadays used in all sectors of activity like livestock, agriculture, etc.

Authors: Thank you dear reviewer. We did it

Line 207:…pregnant women who are allergic to penicillin. 207

Authors: Thank you dear reviewer. We added “are”

Line 208: one study in Cameroon showed that all GBS strains were sensitive to these antibiotics, except one (6%) which was intermediate to erythromycin [9]

Authors: Thank you dear reviewer. We did it

Line 210: and another one in Ethiopia reported 26.5% and 210 21.4% [14]. These reports are in accordance with the results of our study. 2

Authors: Thank you dear reviewer. We did it

Line 232-235: This study showed that pregnancy (P-value= 0.019) and gestational age (earlier gestational age or later gestional age?) (P-value= 0.025) are associated with maternal colonisation.  as reported in other studies, for example in Tunisia only [20], the latter also showed an association with age, which was not the case in our study.

It’s not clear what you are writing here. Is this association between gestational age and GBS colonisation or patient age and GBS? Is it associated with earlier gestational age or younger patients or the other way around? Also as mentioned in your results section it’s not clear which gestational age group are more likely to have GBS. Please clarify.

Authors: Thank you dear reviewer. We revised it. ‘’pregnancy (P-value= 0.019) and earlier gestational age (P-value= 0.025) are associated with GBS infection in women’’

Line 224-226: A study in the United States found that 8% of patients had a positive D-test, indicating inducible resistance to clindamycin [18] and another study in Germany demonstrated 100% resistance to the M-phenotype [19].

None of these references are in your reference list.

Authors: Thank you dear reviewer. We added them

What do you mean by: resistance to the M-phenotype? Please modify your sentence.

Authors: Thank you dear reviewer. We corrected  it

Line 234: Tunisia only [20], the latter also showed an association with age, which was not the case in our study.

It’s not clear what you mean here, please clarify

Authors: Thank you dear reviewer. We did it.  We meant that: A cohort study including 77 pregnant women in Tunisia, reporting a probable strong influence of gestational status on GBS infection and a increased prevalence of GBS carriage from first to third trimester, but unlike us, the authors showed an association with age of patient [20].

Reviewer 2 Report

General comments: The manuscript is short, crisp, written, and presented moderately well. However, a few changes/justifications and details are essential before the manuscript can be considered for publication. The authors should pay attention to the below-mentioned comments to further improve the quality of the work.

Suggested minor revisions:

  1. Consider replacing the word “susceptibility” with “resistivity” in the title to be more specific. The first letter in the word “Women” need not be in uppercase
  2. Mention in the abstract and discussion how fatal this GBS infection can be during pregnancy and in newborns
  3. Mention in the abstract the total number of isolated GBS strains
  4. Abstract: Line no. 20: Specify whether “Yaounde” is a city/state/province
  5. The word “global” in line no. 30 could be misleading. Consider replacing it with “overall”
  6. Justify why genetic identification of GBS isolates using 16S rRNA sequencing has not been made. Why was the study only restricted to microbiological and immunological assays?
  7. Abstract: Line no. 24: Write “collected from” instead of “collected on”
  8. Abstract: Line no. 33: Put the number of antibiotics tested in parenthesis after “all”
  9. Abstract: Line no. 37: “All these indicate” and not “All this indicates”
  10. Line no. 42: Italicize the word “Streptococcus”
  11. Line no. 43: The word “gastro-intestinal” does not contain a hyphen. Make it uniform throughout
  12. Line no. 61: “United States” and not “United states”
  13. The phrase “but reduced to erythromycin” is not clear in line no. 64
  14. Italicize all the bacterial antibiotic resistance genes in line no. 66
  15. Line no. 75: Write “central” instead of “centre”
  16. Provide a brief explanation regarding the second sentence in line no. 80-82 in the “Materials and Methods” section
  17. The phrase “exam realisation” is not clear in line no. 84
  18. Line no. 84: “eligible” and not “elligible” and “study” and not “studay”
  19. Line no. 86: A description of the questionnaire could be added as a supplementary information
  20. Under the subheading “Sample analysis,” make separate paragraphs for each subpoint marked in bold
  21. Provide a reference for using the specific media and CO2 dose in line no. 91-93
  22. Check the word “cj=hocolate” in line no. 92
  23. Line no. 92: Check the spelling of “nystatin.” Mention the reason for nalidixic acid, nystatin, and colistin supplementation.
  24. Line no. 93: “incubated” and not “incubates” and “carbon dioxide” and not “Carbon dioxide”
  25. Line no. 90-114: A schematic flow diagram for the entire “sample analysis” process as an additional figure would be helpful for the readers
  26. Line no. 104: “Resistance” should be in lowercase
  27. Provide a reference for the Kirby Bauer disk diffusion method in line no. 99-101
  28. Line no. 110: “indicative” and not “indicatif”
  29. Line no. 114: Provide a few more references for the L and M phenotypes
  30. Line no. 116: The word “variables” is not clear. Provide a few examples of these variables within parenthesis for a better understanding
  31. Line no. 117: It will be “recorded” instead of “recordes” and “statistical” instead of “statistic”
  32. Line no. 121: “P-value<0.05” and not “P-value<0,05”
  33. Clarify the word “constitutive” as used in the manuscript
  34. Line no. 134-135: Rephrase as “The GBS strains were isolated only from the patients’ vagina using the vaginal secretions”
  35. Line no. 137: The word “non pregnant” should be hyphenated
  36. Subsection 3.1 only talks about vaginal samples. However, the abstract mentioned both vaginal swabs and urine samples. Rectify this disparity by including the urine sample data in the result section with adequate explanations (i.e., in the text)
  37. Line no. 135: The text says “88.66%,” but “84.66%” for vaginal secretions is mentioned in Table 1. Rectify this
  38. Check the word “Effectif” in Table 1. Consider replacing it with “Sample size” or “Sampling population”
  39. Did the authors check the MAR index for the antibiotics used in the study? Please provide that in Table 2 or the result section
  40. Improve the presentation of Figure 1. The word “species” should not be in italics in the figure legend. Italicize “E. coli” and keep it uniform in the figure. Italicize “Staphylococcus.” For some values/percentages, commas should be replaced with decimals
  41. In the methodology, the identification of the other bacteria (apart from GBS) is not clearly mentioned. Clarify with references 
  42. Line no. 146: Write “each.” instead of “each…”
  43. Explain the choice of different antibiotics in the result with respect to Table 2. Check the spellings of the antibiotics mentioned in Table 2
  44. Check the word “Effectifs” in Table 2
  45. Line no. 154: Write “respectively” after “tetracycline”
  46. Check the x-axis of Figure 2 for uppercase and lowercase letters. “Synergies” in the figure title should be in lowercase
  47. Line no. 163: Put the acronym “PBP” within the bracket for “Penicillin Binding Protein” after it
  48. In line no. 183 and 185, “spp.” should be in regular font (and not in italics)
  49. Check line no. 189-191 carefully for authors and years and phrasing
  50. Line no. 194: Write “respectively” after “25.5%”
  51. Line no. 199: The phrase “alert the relevant organisms” is not clear
  52. Mention what “cMLSB” and “iMLSB” mean, as mentioned in line no. 162, 219, 222, and 223
  53. Line no. 222: Write “them” instead of “the” in both places
  54. Rephrase line no. 234-235
  55. Omit the first sentence of the “Conclusions”: “This section is not mandatory but can be added to the manuscript if the discussion is unusually long or complex.”
  56. Line no. 241: Omit the word “antimicrobials”
  57. Line no. 248 and 250: An author’s initial is missing in the phrase “ICD and” 
  58. The language of the “Discussion” section can be further improved. Mention how potent GBS is compared to the other pathogens mentioned in the study
  59. Check for grammatical, punctuation, spacing, spelling, and typographical errors throughout the manuscript
  60. Check if the references are properly formatted according to the journal’s requirement (both within the text and in the reference section)
  61. Keep the similarity index within the threshold limits

Author Response

Dear Reviewer #2

Thank you for your valuable comments, we truly appreciate. Please see our responses below.

Reviewer #2

Comments and Suggestions for Authors

General comments: The manuscript is short, crisp, written, and presented moderately well. However, a few changes/justifications and details are essential before the manuscript can be considered for publication. The authors should pay attention to the below-mentioned comments to further improve the quality of the work.

Suggested minor revisions:

Consider replacing the word “susceptibility” with “resistivity” in the title to be more specific. The first letter in the word “Women” need not be in uppercase

Authors: Thank you dear reviewer. We did it.  

Mention in the abstract and discussion how fatal this GBS infection can be during pregnancy and in newborns

Authors: Thank you dear reviewer. We did it.  

Mention in the abstract the total number of isolated GBS strains

Authors: Thank you dear reviewer.  It is already mentioned in the abstract

Abstract: Line no. 20: Specify whether “Yaounde” is a city/state/province

Authors: Thank you dear reviewer. We did it.  

The word “global” in line no. 30 could be misleading. Consider replacing it with “overall”

Authors: Thank you dear reviewer. We did it.

Justify why genetic identification of GBS isolates using 16S rRNA sequencing has not been made. Why was the study only restricted to microbiological and immunological assays?

Dear reviewer, due to lack of funding the molecular characterisation of isolates in this study couldn’t be made possible. We included it in the new section named “Limit of the study’’. However, the strains are well conserved for eventual molecular tests as soon as funds will be available.

Abstract: Line no. 24: Write “collected from” instead of “collected on”

Authors: Thank you dear reviewer. We did it.

Abstract: Line no. 33: Put the number of antibiotics tested in parenthesis after “all”

Authors: Thank you dear reviewer. We did it.

Abstract: Line no. 37: “All these indicate” and not “All this indicates”

Authors: Thank you dear reviewer. We did it.

Line no. 42: Italicize the word “Streptococcus”

Authors: Thank you dear reviewer. We did it.

Line no. 43: The word “gastro-intestinal” does not contain a hyphen. Make it uniform throughout

Authors: Thank you dear reviewer. We did it.

Line no. 61: “United States” and not “United states”

Authors: Thank you dear reviewer. We did it.

The phrase “but reduced to erythromycin” is not clear in line no. 64

Authors: Thank you dear reviewer. We revised it.

Italicize all the bacterial antibiotic resistance genes in line no. 66

Authors: Thank you dear reviewer. We did it.

Line no. 75: Write “central” instead of “centre”

Authors: Thank you dear reviewer. We did it.

Provide a brief explanation regarding the second sentence in line no. 80-82 in the “Materials and Methods” section

Authors: Thank you dear reviewer. We revised it

The phrase “exam realisation” is not clear in line no. 84

Authors: Thank you dear reviewer. We explained  it

Line no. 84: “eligible” and not “elligible” and “study” and not “studay”

Authors: Thank you dear reviewer. We corrected it

Line no. 86: A description of the questionnaire could be added as a supplementary information

Under the subheading “Sample analysis,” make separate paragraphs for each subpoint marked in bold

Authors: Thank you dear reviewer. We did it

Provide a reference for using the specific media and CO2 dose in line no. 91-93

Authors: Thank you dear reviewer. We did it

Check the word “cj=hocolate” in line no. 92

Authors: Thank you dear reviewer. We did it

Line no. 92: Check the spelling of “nystatin.” Mention the reason for nalidixic acid, nystatin, and colistin supplementation.

 Authors: Thank you dear reviewer. We did it. ‘’nalidixic acid, nystatin, and colistin supplementation was to inhibit the growth of undesirable bacteria (Gram-negative bacteria, other cocci)”.

Line no. 93: “incubated” and not “incubates” and “carbon dioxide” and not “Carbon dioxide”

Authors: Thank you dear reviewer. We did it

Line no. 90-114: A schematic flow diagram for the entire “sample analysis” process as an additional figure would be helpful for the readers

Authors: Thank you dear reviewer.

Line no. 104: “Resistance” should be in lowercase

Authors: Thank you dear reviewer. We did it

Provide a reference for the Kirby Bauer disk diffusion method in line no. 99-101

Authors: Thank you dear reviewer. We did it

Line no. 110: “indicative” and not “indicatif”

Authors: Thank you dear reviewer. We did it

Line no. 114: Provide a few more references for the L and M phenotypes

Authors: Thank you dear reviewer. We did it

Line no. 116: The word “variables” is not clear. Provide a few examples of these variables within parenthesis for a better understanding

Authors: Thank you dear reviewer. We did it

Line no. 117: It will be “recorded” instead of “recordes” and “statistical” instead of “statistic”

Authors: Thank you dear reviewer. We corrected it

Line no. 121: “P-value<0.05” and not “P-value<0,05”

Authors: corrected dear reviewer. Thank you

Clarify the word “constitutive” as used in the manuscript

Authors: Thank you dear reviewer. We did it

Line no. 134-135: Rephrase as “The GBS strains were isolated only from the patients’ vagina using the vaginal secretions”

Authors: Thank you dear reviewer. We rephrased it

Line no. 137: The word “non pregnant” should be hyphenated

Authors: Thank you dear reviewer. We did it

Subsection 3.1 only talks about vaginal samples. However, the abstract mentioned both vaginal swabs and urine samples. Rectify this disparity by including the urine sample data in the result section with adequate explanations (i.e., in the text)

Authors: Thank you dear reviewer. We revised it

Line no. 135: The text says “88.66%,” but “84.66%” for vaginal secretions is mentioned in Table 1. Rectify this

Authors: Thank you dear reviewer. We did it

Check the word “Effectif” in Table 1. Consider replacing it with “Sample size” or “Sampling population”

Authors: Thank you dear reviewer. we replaced it with “Total number” as suggested by the first reviewer

Did the authors check the MAR index for the antibiotics used in the study? Please provide that in Table 2 or the result section

Authors: Thank you dear reviewer. We did it

Improve the presentation of Figure 1. The word “species” should not be in italics in the figure legend. Italicize “E. coli” and keep it uniform in the figure. Italicize “Staphylococcus.” For some values/percentages, commas should be replaced with decimals

Authors: Thank you dear reviewer. We did it

In the methodology, the identification of the other bacteria (apart from GBS) is not clearly mentioned. Clarify with references

Authors: Thank you dear reviewer. We did it

Line no. 146: Write “each.” instead of “each…”

Authors: Thank you dear reviewer. We did it

Explain the choice of different antibiotics in the result with respect to Table 2. Check the spellings of the antibiotics mentioned in Table 2

Authors: Thank you dear reviewer. We did it

Check the word “Effectifs” in Table 2

Authors: Thank you dear reviewer. We revised it

Line no. 154: Write “respectively” after “tetracycline”

Authors: Thank you dear reviewer. We revised it

Check the x-axis of Figure 2 for uppercase and lowercase letters. “Synergies” in the figure title should be in lowercase

Authors: Thank you dear reviewer. We did it

Line no. 163: Put the acronym “PBP” within the bracket for “Penicillin Binding Protein” after it

Authors: Thank you dear reviewer. We did it

In line no. 183 and 185, “spp.” should be in regular font (and not in italics)

Authors: Thank you dear reviewer. We did it

Check line no. 189-191 carefully for authors and years and phrasing

Authors: Thank you dear reviewer. We revised it

Line no. 194: Write “respectively” after “25.5%”

Authors: Thank you dear reviewer. We did it

Line no. 199: The phrase “alert the relevant organisms” is not clear

Authors: Thank you dear reviewer. We did it

Mention what “cMLSB” and “iMLSB” mean, as mentioned in line no. 162, 219, 222, and 223

Authors: Thank you dear reviewer. We did it

Line no. 222: Write “them” instead of “the” in both places

Authors: Thank you dear reviewer. We did it

Rephrase line no. 234-235

Authors: Thank you dear reviewer. We did it

Omit the first sentence of the “Conclusions”: “This section is not mandatory but can be added to the manuscript if the discussion is unusually long or complex.”

Authors: Thank you dear reviewer. We did it

Line no. 241: Omit the word “antimicrobials”

Authors: Thank you dear reviewer. We did it

Line no. 248 and 250: An author’s initial is missing in the phrase “ICD and”

Authors: Thank you dear reviewer. We rectified it

The language of the “Discussion” section can be further improved. Mention how potent GBS is compared to the other pathogens mentioned in the study

Authors: Thank you dear reviewer. We improved the “Discussion’’

Check for grammatical, punctuation, spacing, spelling, and typographical errors throughout the manuscript

Authors: Thank you dear reviewer. We improved all

Check if the references are properly formatted according to the journal’s requirement (both within the text and in the reference section)

Authors: Thank you dear reviewer. Well done as requested

Keep the similarity index within the threshold limits

Authors: Ok dear reviewer. Thanks